# How health securitisation shapes health system priorities: A realist synthesis

**Delaram Akhavein**✪*, **Meru Sheel**✪, **Seye Abimbola**✪

Faculty of Medicine and Health, Sydney School of Public Health, The University of Sydney, Sydney, New South Wales, Australia

* delaram.akhavein@sydney.edu.au

## Abstract

Health securitisation – i.e., framing health issues as security issues or as threats– is growing in contemporary global health discourse. Given the power disparity between the actors who can frame a health issue or on whose behalf issues are framed and the actors who may experience the consequences of such framing, it is crucial to understand how 'health security' framing impacts health system prioritisation. We conducted a realist synthesis on how ("mechanism") and under what conditions ("context") health securitisation impacts health system priorities ("outcome") at different levels of health governance. To capture contemporary analyses, we searched the SCOPUS, MEDLINE, EMBASE, and Global Health databases for articles published from 2010 to 2023. We included 51 articles in the realist synthesis: empirical studies and conceptual analyses on the effects of health securitisation (and associated security-oriented policies, practices and measures), even if implicitly, on health system priorities. Through the synthesis, we identified five *mechanisms* by which 'health security' framing is adopted in response to an event – 'uncertainty' (managing the ambiguity of risks), 'self-protection' (safeguarding own group/national interests), 'self-reliance' (projecting own ability to be autonomous), 'self-preservation' (preserving own institutional credibility), and 'norm-setting' (entrenching long-term policy shifts). We also identified *contextual factors* (e.g., similar past events, inequities between groups/nations, militarisation of governance, donor influence, geopolitical interests) that influence these mechanisms to generate *outcomes* in the form of priorities (e.g., mobilisation of funds, overemphasis on one disease or approach over others, abuse of power that disproportionately impact marginalised individuals, groups, and nations). Each mechanism is influenced by a set of contextual factors which are peculiar to a place and time. Understanding these links between context, mechanism, and outcome can allow for a closer examination and anticipation of 'health security' framing and its impact on prioritisation in different settings globally.

**Data availability statement:** The analysis did not use any additional data beyond what is available in the supplementary document attached as a separate file.

**Funding:** The authors received no specific funding for this work.

**Competing interests:** We have read the journal's policy and the authors of this manuscript have the following competing interests: SA was the Editor in Chief of BMJ Global Health during the period covered by this synthesis and was responsible for handling or overseeing the peer review and editorial process of 3 articles included in this synthesis.

## Key Messages

- Health securitisation is influenced by political, social, and economic context, with decisions often reflecting priorities that may privilege certain populations or regions over others.

- The mechanism through which health security framing is applied (or health issues are securitised) such as uncertainty, self-protection, self-reliance, self-preservation and norm-setting – are dynamic and interrelated and can vary based on the securitising actors' positions, be the governments or international organisations, demonstrating the complex ways in which health emergencies are framed and managed across governance levels.

- Decisions to securitise or de-securitise health responses often hinge on contextual factors like social, political, and economic conditions. In some cases, opting not to securitise can serve as a form of self-protection, sparing entities from complex political implications or resource strain, but may also overlook critical health needs in less prioritised regions.

- Policymakers and implementers must critically engage with and reflect upon the political roots of health securitisation, recognising its potential to disproportionately affect certain groups and their ethical role as 'norm-setters' within a historically colonial framework.

## Introduction

The use of the term 'health security' both in policy and practice has increased dramatically in global health discourse in the past two decades [1,2]. While "security" – as in national security – was once primarily focused on military threats, there is now an increasing emphasis on health-related "threats", especially in the context of infectious disease [3–5]. The inclusion of health within the domain of security has widened and broadened the reach of "security" [6,7]. This process, in which a health threat is elevated to a security concern, is known as securitisation [8]. It invokes a sense of urgency and exceptionalism, which ultimately requires greater attention and resources [6]. Framing health in security terms is thought to give policies and interventions greater traction across myriad political processes that are thought to support how health crises are responded to [8–11]. While it is used within the context of global public health, the primary drivers of such framing are not always those within the health sector nor from places where it is expected to be operationalised, as outbreaks are seen to not only impact population health but directly impact the global economy by disrupting international trade and, by extension, international peace and security [12–14].

However, such health security framing is neither uniformly applied nor universally understood in global public health [6,15]. The intended outcomes—such as increasing political attention, mobilising large-scale resources, fostering international cooperation, and enhancing preparedness [16]—can have varying degrees of success from one country to another, particularly as there is an inherent unequal distribution

of power and resources in the context of global public health [11]. There is extensive critical scholarship on health securitisation, which focuses on the political nature of such framing in health, particularly the notion that Western countries act in their self-interest by framing and securitising health issues, often conflating the Global South as the source of threat [2,6,10–12,17–19]. For example, the conceptualisation of HIV/AIDS focused on its impact on military combat effectiveness [20,21], while the Ebola Outbreak in 2014 focused on its potential threat to regional 'peace and security' (UN Security Council 2014)[22,23], both actions which were supported by the passing of the UN Security Council adopted resolution 1308 and 2177.

Over time, health security has become synonymous with global (public) health, especially in the context of infectious disease outbreaks and emergencies [24–26], with frequent calls from the World Health Organization (WHO) [12,27] and Centre for Disease Control (CDC) [28] to 'strengthen health security'. In the United Kingdom, the UK Health Security Agency replaced Public Health England in April of 2021, at the height of the COVID-19 pandemic [29], all pointing to the normalisation and integration of health security framing within broader global public health systems. Debates on and analyses of health security often focus on the geopolitics of health and security [10,11,24,30], primarily emerging from disciplines such as International Relations [10,13,20] and Security Studies [8,16,31]– which focus on the international and national interface and are written primarily by scholars in the West [6]. Therefore, there is a relative lack of evaluations and analysis on how health security influences how health systems function and how the framing impacts prioritisation by different health system actors at various levels at the country level. When health security is discussed or presented outside its political scope, it is often in a more operational or technical context, such as increased surveillance, preparedness and prevention capacities or the One Health approach [32–35], which are highlighted for their potential to strengthen health security and vice versa [36–38]. In this way, the technical literature overlooks (perhaps justifiably) the varying interpretations of health security, its political roots and the environment in which the framing operates, and the mechanisms through which it functions, which can produce different outcomes in different contexts.

Different contextual conditions – locally, nationally, and internationally – can shape how health securitisation is triggered, operationalised and institutionalised, and its varying outcomes across countries and across different governance levels. Therefore, it is important that health sector actors engage with and understand how deploying the health security framing may impact health systems differently in different environments. In this article, we aimed to contribute to the understanding of the impacts of health securitisation within the complex and adaptive systems that are health systems. Our aim was to develop a framework to explain how, through various causal forces (i.e., mechanisms) that are influenced by different circumstances (i.e., context), health securitisation generates patterns of prioritisation in health systems.

A realist synthesis is particularly suited for this type of analysis as it allows us to examine and uncover how and why health securitisation, or health security practices, produce different effects in different contexts. The realist approach to evidence synthesis is well suited for answering the questions: 'what works, for whom, in what context?' [39,40]. The application of this approach to our synthesis helps to gain a more contingent understanding of how health securitisation unravels on the ground, given the complexity of health systems. Using this approach, we explored how health security functions at different levels of governance – at international, national, and subnational levels. If the underlying rationale for adopting the health security framing and associated actions is primarily to mobilise resources and increase the political importance of the events that triggered it, we aimed to uncover or hypothesise causal links involved in why certain actors choose to securitise the response to an event, how such framing leads to action, and the effects of those actions. The realist approach encourages syntheses of the evidence on complex social realities that go beyond simple or binary effectiveness assessments and takes into account the interplay between structures and agents (systems and individuals) – how the reasoning or rationale behind an action (mechanisms) is influenced by social, political, and economic conditions (context), and, in turn, together, influence the results or impacts of those actions (outcomes) [39,41]. In this synthesis, we aimed to understand how securitisation manifests within different context-mechanism interactions. We aimed to answer– how and under what circumstances does securitisation impact health system priorities?

 

## Methods

### Theoretical framing

For our analysis, we took inspiration from the Copenhagen School's Securitisation Theory, which argues that securitisation is primarily a framing exercise by a certain set of actors (actor 1) to a second group of actors (actor 2) as the audience. It outlines that securitisation is not bound by the use of the word 'security' or the existence of a threat, but rather a political choice that invokes a sense of existential threat [8]. We also followed the work of Balzacq when thinking about the effects of context [42,43]. Balzacq argues that context and the relationship between the securitising actor (actor 1) and their audience (actor 2) impact the success of a securitisation move; he proposes "(i) that an effective securitisation move is audience-centred; (ii) that securitisation is context-dependent; and (iii) that an effective securitisation move is power-laden." Balzacq's emphasis on the audience-context relationship resonates with the realist approach in that it takes into consideration that how securitisation occurs depends on context. We aim to build on that by exploring how the impact of health security framing impact may differ based on the local political landscape, the actors involved, and the priorities of the securitising actor. In either formulation, the frameworks do not explicitly consider a third group of actors (actor 3)—those we theorised as affected by a securitising move: individuals, groups and nations that experience the consequences of a securitisation move but are not the (primary) audience of the move. In our analysis, we explicitly recognised and theorised this third group of actors –individuals, groups, or nations whose realities are shaped by securitisation – to better explain the outcomes of securitisation. See Fig 1, with actor 1 as the securitising actor that typically raises concerns or frames health issues as security threats; actor 2 as the audience of the securitising move; and actor 3 as the individuals, groups, or nations who experience the consequences of a securitising move.

In our analysis, we also sought to add conceptual and practical understandings of the mechanisms through which securitisation operates in global public health that go beyond the focus on geopolitics and surface its outcomes at different levels of governance. We used this framework as a starting point to identify how the interactions of the different actors, depending on the context in which they function, impact outcomes and actor 3. We sought to identify underlying mechanisms – causative factors of the outcome-context interaction – that is, the "generative causes of empirical reality" [41] to provide explanations as to why securitisation was triggered in a particular context (the rationale of actor 1 to use health security framing) and to what end.

In the process of retroductive theorising (Step 3: uncovering hidden mechanisms), we built upon the frameworks. By identifying the underlying mechanisms, we could better understand why certain health events or issues are framed as security threats and how this framing influences the decisions that reshape health system priorities. We identified five mechanisms – 'Uncertainty', 'Self-protection', 'Self-reliance', 'Self-preservation' and 'Norm-setting' (Table 1) and created five sets of Context-Mechanism-Outcome (C-M-O) configurations. Through the analysis, contextual factors in each set were accumulated and linked to their outcome through the mechanisms – with the mechanisms operating in a way that is both reinforcing and overlapping (Fig 3). For example, motivations like 'Self-protection' and 'Self-preservation' may explain why governments prioritise emergency responses over long-term health investments, while 'Norm-setting' could explain the role of global actors in shaping and maintaining these priorities. This understanding helps to reveal how securitisation may lead to or reshape certain health system priorities depending on how these mechanisms interact in different contexts with different actors. Building on the Copenhagen School's Securitisation Theory, we did not distinguish between different security threats and whether real threats exist or not but instead sought to explain the process and the motivations behind health security framing, its consequences, and the responsibility that it carries.

In our analysis, even if a given outcome was not explicitly reported or discussed in a particular setting, but the contextual circumstances and the mechanisms at work in that setting were consistent with those in other settings where the outcome had been reported or discussed, then that outcome could still be abstracted conceptually and theoretically in such a setting. The goal of a realist synthesis is to build on and refine existing theories, allowing for broader insights beyond the

PLOS Global Public Health

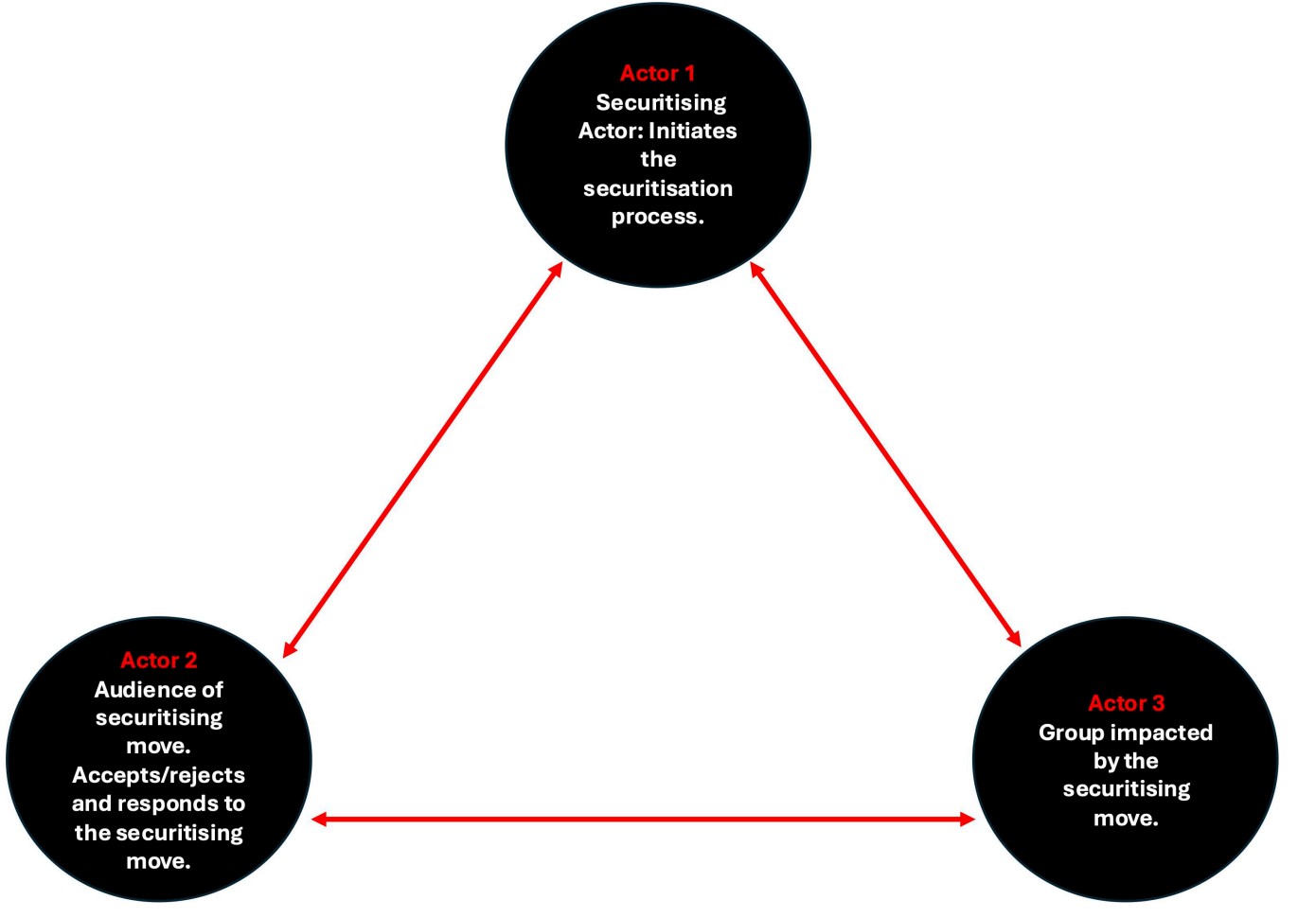

**Fig 1. The Actor 1 – Actor 2 – Actor 3 relationship explaining their securitisation roles and experiences [ 44].**

**Table 1. Identified mechanisms explaining how they lead to securitisation of health and/or use of health security framing.**

**Uncertainty –** resulting from incomplete or conflicting information, especially during disease outbreaks, can trigger securitisation by creating ambiguity about risks to health, the economy, trade, and security, both in new crises and heightened concerns from past events.

**Self-protection –** refers to safeguarding the well-being and interests of a nation-state and its citizens, or in some cases, groups of nation-states or groups within nation-states with more deployable power or leverage. Governments justify the allocation of resources, implementation of policies, and deployment of measures aimed at safeguarding national interests from perceived threats.

**Self-preservation –** refers to (over)asserting governmental authority – as governments try to strengthen and reaffirm their political standing and credibility. This can occur in times of domestic instability and internal political tensions.

**Self-reliance –** refers to governments asserting their legitimacy by demonstrating their capability to address threats to public health and national security. By portraying themselves as proactive and decisive actors in protecting citizens from health risks – allowing them to assert their sovereignty and independence in addressing national challenges governments resist external interference or intervention in domestic affairs.

**Norm-setting –** refers to global agenda-setting to influence country policy and to embrace a norm. Certain individuals, groups or nations work to encourage normative security practices that either intentionally or unintentionally. This mechanism works to enable and encourage nation states to highly securitised measures.

immediate findings and specific settings. As such, in reporting our findings, we did not name specific countries but rather used superscripts to refer to the article where we derived the insights for each part of a C-M-O – see S1 Data. Similarly, in identifying the three groups of actors, not all articles explicitly mentioned or identified the actors; however, with the application of a realist lens, it was possible to abstract and interpret their roles based on the broader context and our theoretical framework. Even when actors were not directly named, their influence, presence, or actions could be inferred from the discussions of power dynamics, decision-making processes, and policy outcomes in a particular setting.

## Search strategy

Between the period of August 2023 and November 2023, we conducted a search of Medline, Embase and Global Health via Ovid, and also of Scopus for articles in the period between 2010 and 2023 (given the recent dominance of the terms health security and health securitisation in global health discourse and the global health literature) using the following terms: (health securit* OR securitisation OR securitization) AND (health policy OR global health) AND (pandemic OR outbreak OR health emergency OR outbreak response) AND (health policy OR global health) AND (surveillance OR data governance OR resource allocation OR quarantine OR border closure) AND (militarisation OR militarization). An initial scoping of the literature was conducted to gain insights into the extent, range and nature of the literature and to support the refinement of the review question. The search terms were identified and pre-tested on Google Scholar, after which the search strategy was developed with the guidance of a research Librarian. The steps and procedures outlined in the RAM-SES publication standards for a realist synthesis were followed in conducting and reporting this realist synthesis [45].

As realist syntheses are data-informed and theory-driven and given our interest in uncovering aspects of health securitisation for which evidence is limited or difficult to generate [41], we included quantitative and qualitative studies as well as non-empirical articles such as conceptual/theoretical analyses and commentaries to provide diverse sources of data. Our initial search sought any studies discussing or assessing the effects of securitisation (conceptually and empirically) and health security policies on health system policies and how they may (even if implicitly) impact prioritisation. Articles were included if they recognised that 'securitisation' was involved in a health issue – that is, if they recognised that the health issue that is the subject of the article was framed or responded to as an existential threat – and explicitly acknowledged and explored the process of that framing or response and its effects in any part of a health system, at any level of governance. This was so that contextual factors could be explored in terms of their effect on outcomes; that is, articles were included if they contributed to any part of the C-M-O configuration – for example, if an article discussed health securitisation as a process, provided contextual insights, or discussed (even if implicitly) potential outcomes. We excluded papers if they did not have a clear focus on 'health security' or 'health securitisation' as central themes, particularly in relation to their potential impact on health systems. For instance, studies that only referred to health programs or interventions for their potential to 'strengthen health security' but did not critically engage with the concept of securitisation as a process— whether positively or negatively—were excluded to maintain relevance to the research question.

## Data extraction and categorisation

The initial search of the databases yielded 429 entries from Medline, 552 entries from Scopus, 736 entries from Embase, and 525 entries from Global Health. The search started by screening the title of each article for relevance. After the initial title screening, DA read the abstract of articles whose titles suggested empirical or conceptual content related to health security practice. After selection based on title and abstract, the entries were reduced to 82 in Medline, 102 in Scopus, 141 in Embase and 48 from Global Health and were subsequently merged with title and abstract screened a second time and duplicates identified, resulting in 97 publications – 1 of which was in Polish and was excluded. After the full-text screening of the 96 papers, an additional 54 papers were excluded based on the inclusion and exclusion criteria and their richness and rigour (DA in consultation with SA). A further 8 papers were found and included through backward citation screening – see Fig 2. The papers were assessed based on two criteria –richness and rigour – the richness assessment

PLOS Global Public Health

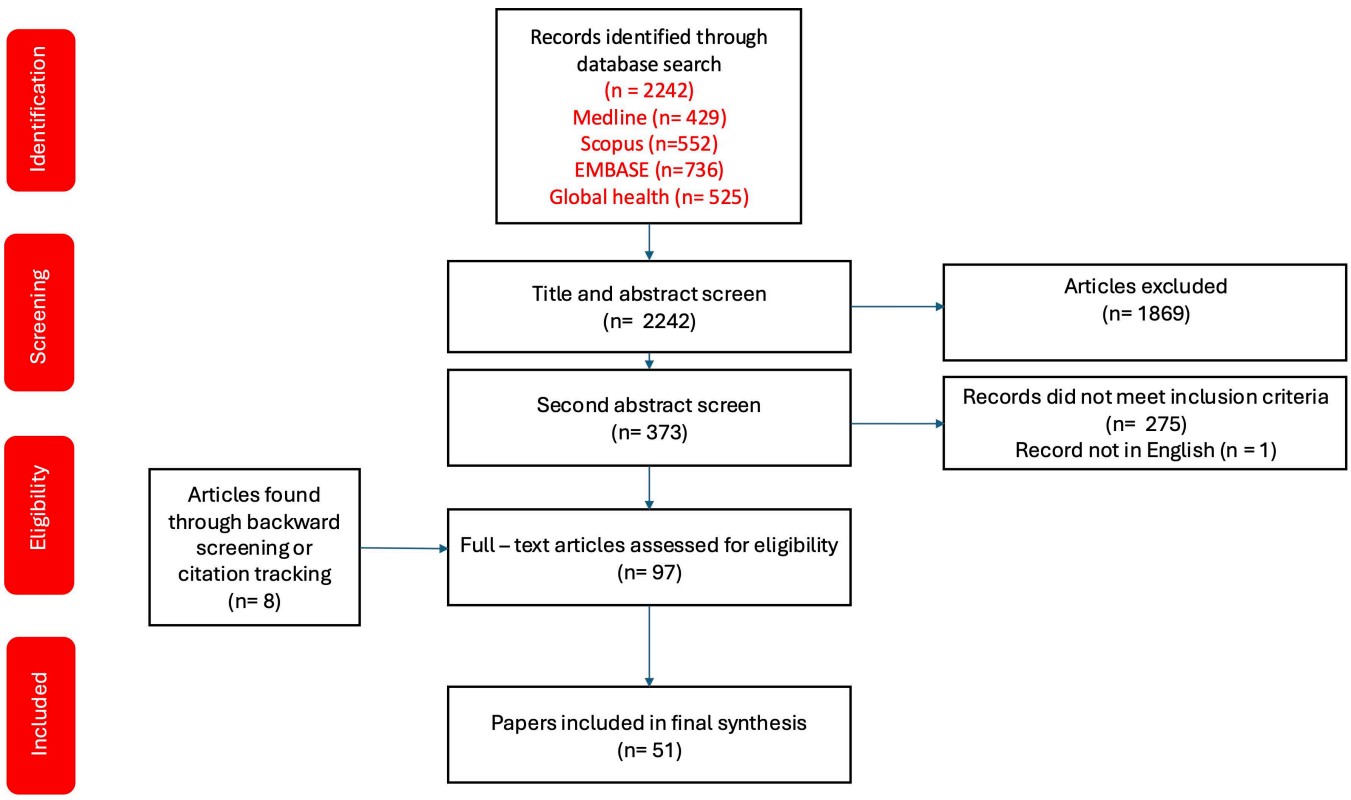

**Fig 2. Document selection flow diagram: the selection of documents for inclusion in this realist review.**

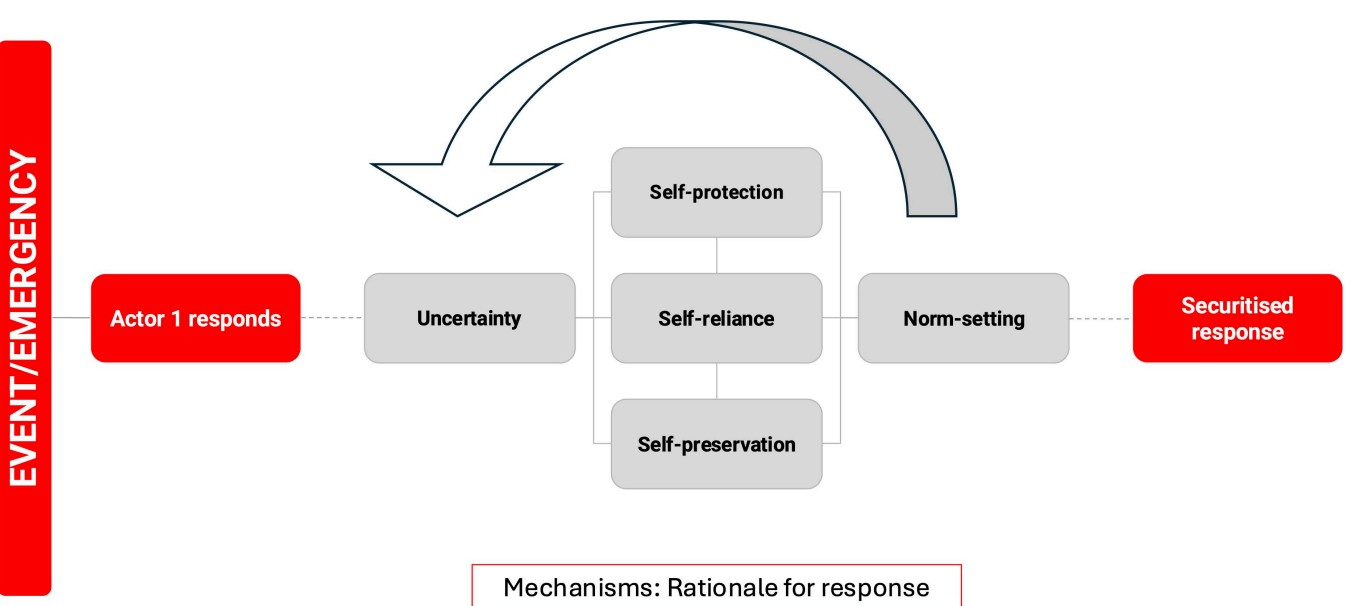

**Fig 3. Pathways to securitised responses: mechanisms by which 'health security' framing is adopted in response to an event.**

was used to assess the depth of insights and was used to determine if any parts of the papers contributed to our causal understanding and theorisation of mechanisms and their impacts on health systems. The rigour assessment was used to determine the credibility and coherence of the data and theories. This is a case-by-case approach and does not involve a formalised appraisal process; instead, it relies on the researcher's reflections and experience by asking questions such as: is the information plausible? Is the information justified? Is the information provided coherent?

We adopted the stepwise approach to realist analysis proposed by Danermark et al. [46]. Data extraction was initially piloted on two papers (DA in consultation with SA). The data extraction was conducted using an Excel data extraction template, which included characteristics such as objectives, setting, type of article, general findings and underlying models/theoretical frameworks used in each paper. The rest of the data extraction form focused on extracting information and evidence, including verbatim extraction of text where relevant to detail a reframing of each paper in terms of contexts, mechanisms and outcomes, as well as the identification of the categories of actors, those initiating the securitisation move (actor 1), those who are the primary audience of the securitisation move (actor 2), and those who experience its consequences (actor 3). These categories and insights were not uniformly or, at times, explicitly reported and were therefore identified from all sections of an article, including the introduction, methods, and results and at times relied on the reflective interpretations of the authors in the discussion or elsewhere in the paper, and conceptual analysis presented in others.

All passages of the papers were searched to identify the actors involved in the securitisation process (see theoretical framing section). In addition, we searched for actions and decisions that may lead to or impede securitisation. We also searched passages of each paper to identify events or the consequences (intended or unintended) of health securitisation, and these were coded as 'outcomes' (Step 1: description). The outcomes were extracted in combination with relational or dynamic features (including but not limited to institutional, political, socio-economic and geographic factors) that enabled or hindered them – and these were labelled as 'context' (Step 2: resolution). The outcome-context sets were tagged with behaviours or invisible rationales that explained them, which were tagged as 'mechanisms'. The list of potential mechanisms was refined through an iterative process until a coherent set of five mechanisms accounted for and explained the effects of securitisation. The outcome-context sets were moved to different mechanisms as more insights were gained and mechanisms finalised.

## Findings

The 51 included publications presented data from 27 countries (Indonesia, Brazil, United Kingdom, Germany, New Zealand, Uganda, Ethiopia, Sierra Leone, United States, Canada, Australia, China, Liberia, Guinea, Israel, Russia, Pakistan, Malaysia, Vietnam, El Salvador, Colombia, Zimbabwe, Panama, Colombia, Greece, Tanzania and the Philippines) were represented in this synthesis, with high, middle and low-income and low-income countries. We identified five mechanisms that trigger securitisation: 'uncertainty', 'self-protection', 'self-reliance', 'self-preservation', and 'norm-setting'. We did not develop the C-M-O configurations based on specific countries but rather used insights from all the articles to capture the C-M-O's (see Fig 4). Neither did we take an entire country as "the" context, but rather, we identified the granular circumstances within a country (i.e., the social, economic, and political circumstances). The outcomes in each configuration were identified through the synthesis of the articles.

## Uncertainty

As a mechanism that triggers securitisation, uncertainty may arise from incomplete or conflicting information about a potential risk, particularly during infectious disease outbreaks, creating ambiguity about the risks' nature, scope, or severity. The uncertainty may be related to a risk to health, local economy, international trade and national security. Uncertainty can work reactively during times of 'new' unprecedented events and proactively when a previous emergency or crisis heightens uncertainties about future scenarios, triggering another mechanism, "norm-setting" (see below), which may

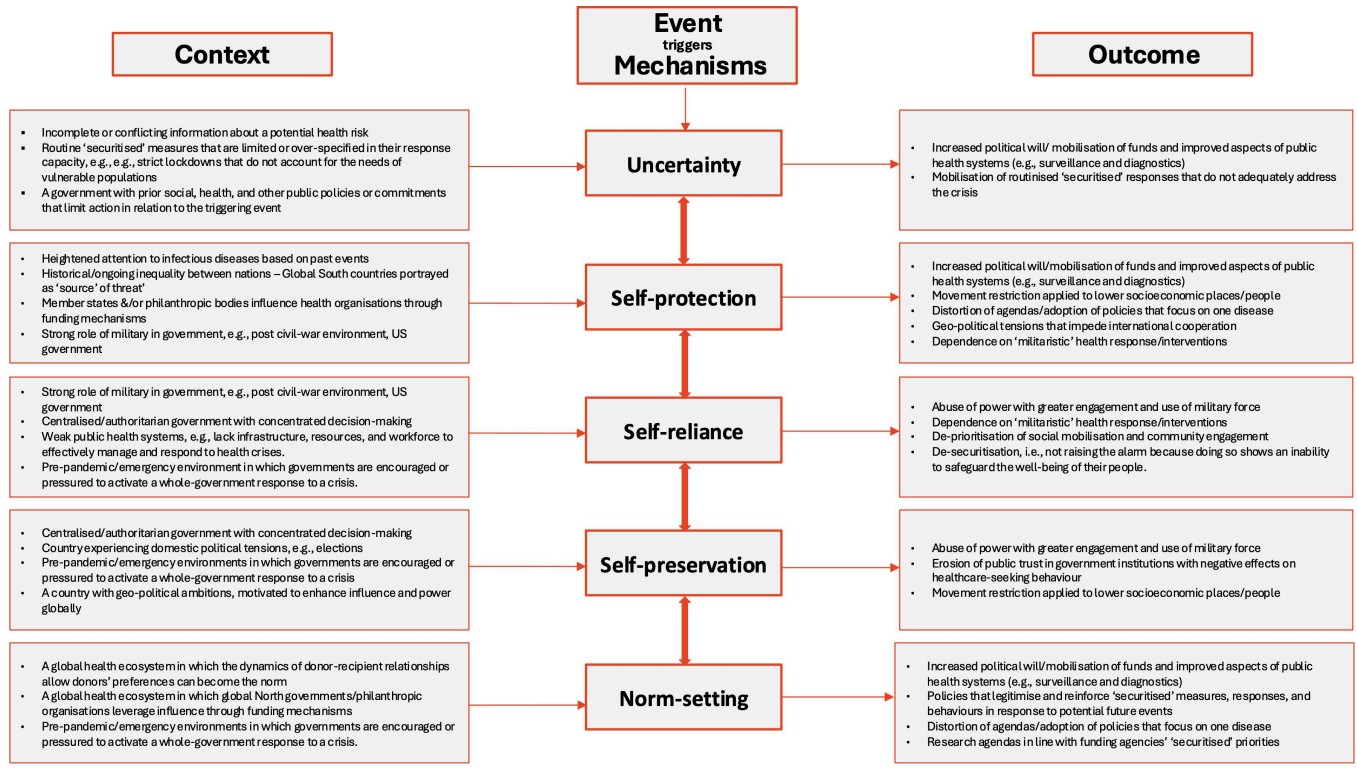

**Fig 4. The Context-Mechanism-Outcome (C-M-O) configurations explain how the identified mechanisms in certain contexts lead to specific outcomes.**

lead to long-term, embedded securitised policies. In the context of an outbreak with unknown pathogenic potential, such uncertainty may inform the steps taken towards securitisation [1,2,3] such that to react to a crisis, WHO (actor 1) may declare a Public Health Emergency of International Concern (PHEIC), with "the decision… [to make such a declaration] not made based on what is currently known, but was rather made based on what is not known" [1].Such uncertainty-triggered actions, including by governments, may place a "heavy emphasis on the international spread ('exportation')" [4] and focus on "global health security responses: containment, surveillance, and detection" [4]. While these measures and declarations are seen to be deployed to protect public health, the sense of uncertainty actions can also encourage some governments to adopt additional securitised measures as a precaution against other potential, unknown effects[10] and depending on the context in which additional measures are operationalised it can lead to varied and complex outcomes.

For instance, uncertainty-triggered securitisation by governments can lead to policies that target specific groups, as in the case of an infectious disease that predominantly affects pregnant women and their newborns (actors 3), leading to a situation in which "the recommendation was not to get pregnant, nothing more" [5], and in which "if women do appear in emergency rooms [having sought termination], we see them handcuffed" [5]. The fear of unsafe pregnancies may prompt some women to seek unsafe or 'illegal' abortions, with health risks and legal ramifications. In conservative social and political settings, such securitised responses may be further conflated with restrictive abortion laws when the burden of preventing pregnancies is solely put upon women[1], neglecting broader issues of reproductive health and autonomy. This outcome may be further exacerbated by socioeconomic disparities, another contextual enabler, with marginalised women (actor 3) facing additional barriers to accessing (safe) reproductive healthcare – while upper and middle-class women are better able to escape the securitised policy by being better able to leverage their status and wealth to exercise otherwise

constrained reproductive rights [1,5]. Also enabled in conservative social and political settings, securitised responses may take the form of sex-segregated lockdowns to suppress the mixing of genders and reduce the chances of transmission in the population [6], which, without gender identity laws, can significantly impact transgender people (actor 3) who may be unable to provide ID (identity) cards with their preferred gender identity, which can expose them to discrimination by the police [6]. There can also be broader implications for other marginalised groups (actor 3), such as those with disabilities who rely on caregivers of the opposite sex[6]. Individuals' access to healthcare may also be impeded if, for example, appointments are not provided on their assigned days [6].

Uncertainty-triggered securitisation by governments and international actors – in relation to an outbreak's future impact and potential return – can also manifest as increased funding and resource mobilisation to the issue of concern relative to other issues. Such optimisation of capacities during or in the immediate aftermath of a crisis – in the form of increased surveillance, diagnostic capacity, and production of healthcare workers – although beneficial, may not be sustainable in the long-term, and can damage the healthcare system [7,8] by enforcing longer-term increased mobilisation of funds and other resources [9] for one infectious disease. This 'over-optimisation' can sit in contrast with attention given to non-communicable diseases: "In 2013, low-income countries with a high burden of HIV/AIDS earned 200 times more disease-specific spending than a low-income country with a high burden of non-communicable diseases" [10] – with potential effects on vulnerable or marginalised populations if specific interventions and capacities are overly tailored to achieve a specific target, rather than universal access. In the short or long term, reactive response can also lead to 'over-optimisation' of – at times, vertical – systems that focus on specific technical capacities[11] without the flexibility and adaptability necessary in low-resource settings. Additionally, responses driven and predominantly funded by donor agencies and donor countries may have a latent impact such that they "affect the way that resources are later allocated to… 'threats' that are measurable and quantifiable – primarily as 'evidence campaigns' for the benefit of donors" as well as the benefit of the national government if occurring sub-nationally [12] thus creating an overemphasis on metrics and data that can lead to bureaucratic overload and diverting from direct patient care.

In some cases, uncertainty may activate or operate as a precursor to other mechanisms: "self-preservation", "self-protection", and "self-reliance" (see below). For instance, if organisations or governments choose to securitise because of the uncertainty of a potential crisis, they may also be doing so as acts of self-preservation to reinforce their credibility [1], the need to appear self-reliant in the face of an unknown crisis[2] or to self-protect from the 'other' [7,8].

## Self-protection

Self-protection as a mechanism that triggers securitised measures manifests when national or international actors frame health issues as matters of national security. This rhetorical move may work to justify the allocation of resources, development of policies, and use of measures deemed necessary to address the relevant health issue. While the mechanism can (implicitly) protect the public within a nation-state, protecting the public for the sake of public health is not necessarily the driver. The measures that stem from self-protection are designed to safeguard a nation from perceived threats to the well-being and stability of a nation-state as a political entity. Policies motivated by this mechanism, such as heightened surveillance of specific diseases, border control measures and quarantine protocols, can disproportionately affect certain 'othered' countries and groups (actor 3) negatively, including migrants, refugees, and other marginalised communities. This mechanism can also worsen existing inequities and power asymmetries between nation-states, especially as self-protection works within the framework of historical and ongoing inequality between nation-states, such that certain countries, especially LMICs (actor 3), are stigmatised and framed as sources of threat [7,11–16]. The rationale for the self-protection mechanism is the presumption that these countries face challenges in effectively monitoring and managing outbreaks within their borders, and therefore, certain policies are required to counter such 'incompetency' [17].

Enabled by a heightened sense of threat to a nation-state posed by infectious diseases and legitimised through United Nations security resolutions (formally) and through all-encompassing governmental response frameworks

(informally)[14,20,23], securitisation triggered by self-protection can generate an 'all-government' response, including military participation in domestic and international emergency response[1,12,14,18,19]. Military participation can manifest through the deployment of military units from Western countries to affected areas primarily to safeguard their citizens and contain the outbreak at its origin[14,20,21]. The impacts of such a deployment can be conflated in a country with a recent history of civil conflict or in an area with civil unrest within a country[20,22] and, as a result, greater military capability than health system capacity, leading to an over-dependence on militaristic approaches to health crises – "the military is all we have" [14]. In the immediacy of civil unrest or in the aftermath of a civil war, with community mistrust of the government given prior coercive and punitive actions, using forceful measures for compliance and to manage public health crises intensifies mistrust[14]. The perceived threat posed by security forces and fears of repercussions on individuals' agencies and livelihoods can lead to resistance from community members [14,24,25] or violence against health workers who are perceived as agents of the government's coercion[14]. This may, in turn, lead to dependence on force to maintain control and the eventual withdrawal or extraction of foreign response workers[24]. Such a chain of events can worsen the relationship between governments and communities, impede the potential for effective public health measures in the immediate and long-term, and profoundly influence healthcare-seeking behaviour within communities – meaning that those who are worse off continue to suffer (actor 3).

Self-protection-triggered securitised measures during disease outbreaks with the potential for transnational spread to the West (actor 1) can manifest in actions to protect Western countries (and, by proxy, their people) from the source—the country from which an outbreak originated (actor 3)[13,14,18,26]. Health security measures such as strict border controls and travel bans are prioritised [3,17], and while they mitigate the spread of diseases, they are driven by socio-political motivations and historically biased assumptions rather than concerns for the health of affected populations [7,13,14]. Self-protection can generate resistance rather than cooperation by nations that have grown increasingly more aware that the response of Western countries selectively prioritising the health of certain groups or nations [27,29,30], thus creating mistrust which can manifest in reluctance to cooperate with the international community during times of crisis due to a perceived lack of mutual benefits, further escalating political tensions [30,31,32]. The erosion of trust between nations, reluctance to engage, or breakdown in international cooperation can impact health prioritisation efforts, impede effective resource allocation and coordinating efforts, hinder collaborative responses, and undermine multilateral initiatives. It may also lead to the adoption of unilateral approaches, widening the divisions and hindering collective efforts to strengthen health systems and address health inequities globally [17,29,30]. Self-protection can lead to countries' deciding to prioritise new surveillance systems and data management practices, accompanied by "biomedical reductionism" [11,16,27] and to prioritise short-term, disease-specific solutions that do not address the underlying causes of potential diseases [10,11,13]. This pattern of prioritisation may be at the expense of other health system needs as it mobilises in-country resources and attracts resources from international development agencies [8,10,13]. This dynamic is exacerbated in health systems that rely on international funding or countries where donor organisations wield significant influence such that "financial support to the health sector is conditional"[8] on prioritising health security initiatives. This dynamic can shape and influence a country's health policies if it leads to a policy loop in which past securitisation efforts continue to shape future public health responses through the lens of health security [7,12,28].

Self-protection can also infuse domestic practices when the health security practices it triggers combine with a generally securitised environment in response to other crises [2,24] or are applied as part of a 'global response' [12]. In such situations, communities such as those in lower socioeconomic areas (actor 3), may be overpoliced as a potential source of threat and as those who need to be "controlled for the sake of their own health" [17,18]. Over-surveillance in relation to certain diseases of pandemic potential (referred to by WHO as disease X)[47] – which may be a result of health security practices – can operate at national borders to limit certain migrant populations (actor 3) from entry or settling [33]. This outcome of self-protection may be enabled by governments enacting new legislation to protect their 'body politic' from a 'migrant' disease [33], such that members of marginalised groups who are infected with disease X (actor 3) may be discouraged

from seeking settlement or healthcare services for fear of detainment and, at times, deportation post-settlement. The self-protection framing of a Western nation "at war" [4,24,26,43] against an existential threat across the border can perpetuate a conception of the 'we' – a white nation that requires protection –and may exacerbate the othering of other nations but also of racialised groups within such a Western nation [12,38]. This can create distrust within the public for people considered as being from the 'outside', including racialised groups who live in the supposed 'white' nations, systematically pushing the racialised group to the margins of society, reinforcing social hierarchies, and perpetuating discrimination and exclusion by the public and law enforcement in the form of increased surveillance and application of restrictive immigration policies [15,18,33,38].

## Self-reliance

Self-reliance as a mechanism reflects national governments' efforts to show – without external interference – their ability, authority, and effectiveness in handling 'national threats' and their capacity to safeguard their citizens, thus enhancing their domestic and international legitimacy. Self-reliance may also manifest as the need to invest in strengthening health-care infrastructure, developing research capabilities, and diversifying sources of funding for public health initiatives. In settings where national governments have either been portrayed as a source of threat or are experiencing geopolitical tensions with Western countries, self-reliance may manifest as a form of 'sovereign posturing' or projection of 'in-house' capacity to respond [19,27,30,34]. With self-reliance, during an active emergency, governments are motivated to increase health funding to increase emergency surge capacity or use domestic funds to secure medical supplies and treatments such as vaccines and can operate vigorously to control the response [34,35]. With self-reliance, national governments move from being the audience of securitisation (actor 2) to securitising actors (actor 1) themselves, especially given "pre-emergency" global discussions on the need for countries to have preparedness and response mechanisms in place [12,37]. With self-reliance, an actor – such as a national government – may moderate its response to an emergency or crisis and 'de-securitise' an issue by withholding information about a potential outbreak from the public but also from the international community to have a chance to respond internally, coupled with the notion that not raising the alarm shows an inability to safeguard the well-being of their people [30]. A self-reliance-triggered response may also construct an 'us against the rest' dynamic by the government – " we do not give in to fear" and "we overcome, and we win" [38] – the word 'we' is thus used to bridge the gap between itself and the public while also creating a distance between itself and the 'external' community [31].

Self-reliance-triggered securitised measures manifest as vigorous efforts to ensure compliance and control an out-break, which may fail to consider marginalised groups, such as those living in high-density areas or those whose liveli-hoods depend on informal incomes [24,25,36]. Governments at national and subnational levels may prioritise resources and funds to arrest individuals for non-compliance or "demolish slums" as response mechanisms rather than prioritise patient care and access to healthcare services [12,19,25]. For instance, in a self-reliance environment in which countries are driven by the need to 'eradicate' a disease, marginalised groups, such as refugee populations (actor 3), can be seen as limiting factors in achieving such goals – "non-immunised refugee children are constructed not merely as public health problems but simultaneously labelled insurgent "national security" threats" [13]. These vulnerable populations may bear the brunt of heightened surveillance and suspicion – the securitisation of marginalised groups of the population can result in stigma-tisation, a direct risk to their health, and can have severe consequences for how they may or may not seek healthcare in the future [25].

In areas with civil unrest or in a post-civil war setting that has continued to fund its military capabilities and not its health system, there can be consequences (for actor 3) [20] when self-reliance manifests as over-reliance on the military and coercive actions by the government (actor 1) to enforce compliance with securitised practices, and as a de-prioritisation of social mobilisation and community engagement [22,24,25,34]. When governments prioritise lockdowns enforced by the military to assuage or counteract limited health response capacities, they may not take into consideration impoverished and marginalised communities (actor 3) whose livelihoods are predominantly through informal jobs. Even if there were

governmental subsidies (often not the case) for loss of income, the informal job keepers would not be eligible [24]. This can exacerbate socioeconomic inequalities, pushing marginalised groups further into poverty and increasing their susceptibility to disease [16]. Informal workers, urban poor, and women [39] can be made more vulnerable to exploitation and abuse – "police officers sexually exploiting women in exchange for a quarantine checkpoint permit" [24,] which undermines their dignity and rights, perpetuating cycles of inequality and injustice. In decentralised settings, subnational governments, either away from national government scrutiny or to exercise their autonomy, can create additional – at times unnecessary- measures that can create an authoritarian rule as a means of control. There may also be "inequitable enforcement"[25] of securitised measures when government officials or members of elite groups are treated far more leniently for violating rules than the less privileged groups [5,14,37]. This may extend to access to life-saving health services, with disparities between who is cared for in hospitals (secure) and who is left to fend for themselves (insecure). Such practices – whether intentional or not – may increase mistrust between the government and the communities, who feel unprotected during times of crisis.

**Self-preservation**

Self-preservation as a mechanism reflects governments' efforts to (over)assert their authority and strengthen their political standing and credibility, nationally and internationally. Seen in contexts of domestic instability, political unrest, or challenges to their legitimacy, governments can present themselves as proactive, capable of addressing health risks, competent and suitable to govern during a health emergency, which can also serve as a distraction from internal issues or failures of governance. By framing health issues as urgent and existential threats, governments may justify restrictive measures and curtail civil liberties for self-serving reasons in the name of public protection. The portrayal of governments as protectors – with "'performative enactments of national manhood in its sovereign mode" [38] may relegate the public to subordinates who should embrace the government's practices [39]. These practices may resonate with groups likely to be less impacted by their effects – particularly when such groups "judge that the directions of authority are morally appropriate and enforced in ways that are fair" [42]. Governments may become reliant on this group to enforce measures amongst the population themselves, therefore enabling them to normalise securitised practices. A government's self-preservation can also manifest as 'de-securitisation' [30,44], minimising or concealing the severity of an outbreak or, pressured by the need to avoid global scrutiny, refraining from full cooperation with international bodies like WHO [27,29,30]. In regimes with a history of authoritarianism, the public may perceive this lack of action from 'de-securitisation' as a form of neglect, but when groups of the population challenge the government's handling of health crises, protests can be met with repression, exacerbating tensions and further eroding trust [19,30].

Enabled by a contentious political climate, self-preservation makes governments securitise their nations during public health crises for potential self-gains: "[The President] is using corona for doing his own things"; "the government is using corona to enrich itself..." [18,22,25,40]. Governments may use elements of securitised responses to emergencies, such as lockdowns and surveillance, as a means of controlling the population during times of governmental (re) elections by diverting resources away from essential public health resources – "state diverted its attention and resources from mitigating the COVID-19 pandemic to quelling protests and insurgencies" [22]- towards self-serving activities that can hinder the effectiveness of any response effort and prolong the duration and severity of the crisis. Such examples also demonstrate that governments' prioritisation may not be in total alignment with having the population's health in mind, "It is evident that the State of Emergency has been used to secure politics rather than preventing the spread of the virus" [22]. Over time, such actions can erode public trust in government institutions [22], undermine adherence to public health guidelines [16,19,41], and exacerbate health disparities in the population [5,6,24]. In highly militarised settings, securitised practices triggered by self-preservation can breed corruption – "soldiers often receiving payment directly to their phones in advance of their arrival at a roadblock" [19] – and abuse of power to enforce compliance: "fired live rounds of ammunition into the air and beat traders until they ran away"[19]. There may also be a latent effect of the increased involvement of the military in future emergencies – "positions usually held by civilians were filled by retired military and former police officers"[24] – further

legitimising their involvement during and after an emergency in the form of increased funding and direct involvement in handling public health crisis [2,13,20,25].

Self-preservation measures can also extend beyond individual nations' borders to reflect their need to safeguard their broader geopolitical ambitions – which may overlap with the need to safeguard their own population to 'self-protect' [32] – as particularly evident in health security strategies developed by Western countries. For example, mobilising resources and expertise in health through partner programs and contributions to international health organisations may be beneficial to recipient countries and project an image of benevolence but be indicative of a preoccupation with using global health efforts as a tool for influence and diplomatic standing[42,45], which can also shape how funds and resources are allocated and prioritised.

### Norm-setting

Norm setting as a mechanism refers to national and international actors' agenda-setting, shaping and operationalising securitised policies and reinforcing the previous mechanisms. Norm-setting occurs as an autonomous act of establishing new agendas and may also be enacted in compliance (norm-compliance) with or in response to a previously set (health security) agenda or norm. In this sense, norm-setting can be shaped by external influences, where actors comply with pre-established norms – which may be set elsewhere – embedding and institutionalising those norms in their own setting. [48]. Even when it does not directly promote 'negative' practices or outcomes, it remains a mechanism through which health security agendas are reinforced and localised. Norm setting operates through institutional socialisation, where actors work to socialise other actors into accepting and embracing health-related security norms. This can happen vertically (that is, the normalisation of practices from the West), and horizontally (that is, across and within governments). However, the influence of normative health security practices extends beyond governmental policies and can also permeate academia and shape research agendas. "Publications in the PubMed database that mention both 'health' and 'security' in the title or abstract have risen exponentially between 1980 and 2022" [49]. This directly reinforces health security norms and narratives and uncritical adoption of its impacts not otherwise assessed or widely discussed [42] – "These publications raise issues and reshape intellectual agendas; they seek to extend or reshape the mandate of particular organizations; they raise the policy profile of organizations" [50].

Vertical norm-setting involves influencing countries' policies to adopt norms related to security practices [9,45–47]. It is operationalised as an extension of 'global health'; that is, public health in lower-income countries- where powerful individuals, groups, or nations work to promote certain normative practices. It influences and may control state capacity and response to outbreaks, especially those with the potential for transborder spread. Such a way of operation solidifies and legitimises the security framing of health [12,23,46] - "threat construction may be accepted and become institutionalised in policies, practices and logics which eventually become part of a new normality" [28]. Vertical norm setting can manifest as international health organisations' declarations, such as the WHO declaring Public Health Emergencies of International Concern (PHEIC) [1,2,47,48] – and, by extension, paving the way in which countries may respond to the alert - "even though it is not backed up by direct enforcement capacities, IO [International Organisation] decisionism sets the agenda for state behaviour in times of crisis, as well as for further securitizations" [45]. Beyond acute securitisation, that is, during and in the immediate aftermath of a crisis, health security framing may extend, unquestioned, to routinised practices. Such extension of health security practices may occur through compliance with International Health Regulations (IHR), and may normalise a state of crisis that may prevent engagement with structural issues: "health security often prevents or, at least, postpones the necessary debate about social, economic, and political determinants of health" [11]. The implementation of IHR is instituionalised through assessment and implementation tools such as the Joint External Evaluation (JEE) process, the Self-Assessment Annual Report (SPAR) as well as the National Action Plan for Health Security (NAPHS). These tools link powers to domestic legal and administrative processes by encouraging national governments to take action, by monitoring and evaluating national 'health security capacities', their emphasis on technical compliance, and in turn, norm-setting

may reinforce securitised approaches without addressing underlying health inequities. Vertical norm-setting can be further reinforced when 'philanthropic' organisations such as the Bill and Melinda Gates Foundation provide funds to support said evaluation processes and, in turn, contribute to the securitisation process [16,46]. Coupled with conditional forms of funding - "you go along with it if you want to get funded" [13], these enforcements and engagements with the health security framing and practice result in the increasing need by some nation-states and organisations to uncritically 'comply' [39]- "a process of international socialisation whereby states conform to a type of "peer pressure", adopting the new norm as it is perceived to be in their interests to do so. Esteem, arguably, encourages norm compliance" [47].

Horizontal norm-setting is enabled in settings where the securitisation processes were triggered by vertical norm-setting are not attentive to the reality that "understanding the domestic interests of the party in power in nondemocratic states is crucial to understanding how a government will react to health crises" [32]. In such settings, norm-setting may reinforce or normalise the need to respond and detect through a 'whole-government' approach. Under the banner of 'whole-government' involvement, in contexts where health services are lacking, the emphasis on security sector involvement in disease response may lead to increased military engagement and funding [12,13,34,39,51]. Such promotion of security-sector engagement "may heighten insecurity for women and other marginalised groups" [39], disconnecting health policies from community needs and prioritising a 'global' response rather than one that embraces the specific needs of the community in their context. Pre-existing domestic norms also influence how health security norms are interpreted. The normalisation of military involvement at the national level, whereby the government aims to centralise its power, can trickle down to lower levels of governance at the sub-national level, especially when autonomous sub-national governments exhibit ideologies similar to those of the central government [24,25]. For instance, if a government rules from a place of coercive populist leadership, they may choose to 'tackle' a crisis that's not health-related predominantly with the use of force, police involvement and leverage their military capacity [25,43]. Even for a health issue, an effort to comply, appear self-reliant, and preserve their legitimacy, governments may act such that "provincial officials, in a moment of mounting international pressure, called upon armed intervention to reinforce a narrative of progress" [13]. Horizontal norm setting may thus manifest in the form of a securitised policy environment that provides a government with the precedent and opportunity to employ similar methods in future events, regardless of their justifiability, and by extension, creating a norm. When such security practices become ingrained in public health responses, governments at the sub-national level, for example, in compliance with the national government norm-setting (often permeated from international standards) often allocate significant resources and attention to addressing 'health security' concerns, sometimes at the expense of addressing the needs of their population [14,20,34] that can differ to the priorities of the national government and international organisations, particularly those already marginalised or underserved. Investing in reactive securitised and reactive measures instead of underlying health disparities or proactive healthcare strategies can further entrench such a cycle.

Norm-setting – whether vertical or horizontal – also triggers outcomes associated with policies and compliance measures that may be perceived as positive, such as increased diagnostic, surveillance, and quarantine capacities and international cooperation [7,35,39]. The tendency for such capacities to be developed reactively and in silos, unintegrated with other priorities post-crisis, can create a latent burden for health systems such that it "inscribes a very different politics into the landscape of global health that limits the long-term sustainability and inclusivity of vertical disease interventions." [13]. It may also do so at the risk of relegating poorer countries to the role of early warning "canaries" [12] for pandemics that receive support to improve "health security" that is not only inadequate to address the broader health system's needs [42,51] but also does not prioritise marginalised groups. They remain vulnerable to recurrent health crises, trapped in a cycle of reactive responses rather than proactive, sustainable solutions.

## Discussion

Health securitisation, the framing of health issues as security concerns during emergencies and their integration into broader health systems, may be driven by five generative mechanisms: 'uncertainty', 'self-protection', 'self-reliance',

'self-preservation', and 'norm-setting', which can be overlapping and reinforcing. These mechanisms are shaped by various political, institutional, geographic, and socio-economic contexts, which either facilitate or hinder their activation and inform their manifestation in the form of health system prioritisation - see Fig 4 for a summary of the C-M-O configurations. At the heart of the synthesis is that the mechanisms are operating at different levels of governance by different actors (actor 1) and intermediaries (actor 2) within a complex global public health governance system that is often operating in the shadows of politically driven motivations with real consequences for certain groups and nations (actor 3).

In line with Copenhagen School's Securitisation Theory, the aim of this review was not to determine whether infectious disease outbreaks or the lack of specific public health capacities are "really" a security threat but rather provide an analytical tool with empirical evidence "to ask with some force whether it is a good idea to make this issue a security issue – or to transfer to the agenda of panic politics". The mechanism of 'uncertainty' can function as the first step in the operationalisation of 'panic politics' as multiple actors (i.e., international organisations, nation-states, and national and subnational governments) respond to emergencies based on the unknown. The mechanism of uncertainty operates as more than just a reactive response to health crises; it reveals how responding with this rationale can reshape governance structures, priorities, and power dynamics. When uncertainty triggers securitisation, it does more than mobilise immediate public health measures—paradoxically, it can introduce a mindset of risk aversion that permeates decision-making, leading to prioritisation of protective and, at times, exclusionary measures that go beyond the involvement of the health sector, for example, the participation of the military in health response [49,50]. Therefore, the securitisation of uncertainty can become a tool through which authority is asserted, populations and responses are controlled, and resources are diverted, with consequences that extend well beyond the initial crisis.

The mechanism of 'uncertainty' can be triggered and function in isolation, and it is particularly pertinent in the context of emerging infectious diseases such as COVID-19, or in the context of re-emerging diseases in new geographies. In such scenarios, limited knowledge about transmission dynamics, severity, or scale can prompt rapid responses driven more by anticipation and fear. As a result, 'uncertainty' becomes a powerful mechanism that can shape securitised responses from the outset of an outbreak. However, our findings suggest that it can also operate as a precursor to the need to 'self-protect' or to protect the state from perceived external threats, which may then result in the exclusion or marginalisation of certain groups—migrants, refugees, and other vulnerable groups—which are framed as 'others' and 'unknown' carriers of the risk.

More broadly, securitisation does not appear to require the presence of an objective or immediate threat; it can be activated by the *perception* of future instability or risk. For example, during the 2014 Ebola outbreak in West Africa, international involvement was not only driven by humanitarian concern but also by strategic interests. From a 'Western perspective', military and international assistance were perceived to be necessary to maintain political and economic stability, and to avoid potential state failure or regional conflict, including the perceived risk of mass migration to the West [51]. In such cases, securitisation functions as a mobilising force and a mechanism of self-protection, linking distant threats to domestic concerns [52].

While 'self-protection' is concerned with defending the state, 'self-preservation' focuses on maintaining the credibility and legitimacy of institutions, governments, or organisations. The mechanism of 'self-preservation' can be activated when sub-national, national, or international entities seek to protect their authority or reputation during health crises. The declaration of PHEICs, for example, can be as much an act of 'self-preservation' as it is an act of 'self-protection' –the prioritisation and the protection of the West is often made apparent through the timing of when a disease is declared a PHEIC; that is, linked "with the onset of health risks within the Global North" [53]. Subnational governments that are aware of processes that can garner national government support can also deploy a local tool, e.g., 'state of calamity' [54], to trigger a response or support from their national counterparts, out of fear or uncertainty of potential burden on their healthcare system.

The mechanism of 'self-reliance' allows us to consider the inner workings of nation-states compounded by the need to 'self-protect'. It can, for example, manifest when a state or institution acts in ways that reinforce its capacity to deal with

health threats independently, often as a means of avoiding reliance on external actors or international aid, which can be triggered by geopolitical tensions and, in the context of historical inequalities [55,56]. Conversely, in the case of 'vaccine hoarding' during the COVID-19[57,58] pandemic, countries demonstrated 'self-reliance' within a securitised environment by prioritising vaccine production and procurement while engaging in acts of 'self-protection' at the cost of inequitable access for many countries in the Global South while securing domestic populations and national interests.

The notion of 'de-securitisation' that occurs through the mechanism of 'self-preservation' and 'self-protection' can also conceptually occur as an act of 'self-protection'. Take the case of mpox, which in May 2022 was perceived as having been declared a PHEIC only after cases emerged outside African countries [53]; subsequently, it was no longer considered an international emergency once cases were falling primarily in Europe and North America the following year in 2023, while cases were ongoing in several African nations – raising the question of whom the PHEIC measure primarily aims to protect when operationalised at the global level [6,59]. The declaration of another PHEIC in August 2024 was in response to a new strain and the 'uncertainty' of potential re-spread globally. This form of operationalising such public health tools, wherein the "focus is placed not on actual diseases but more precisely on 'events'" [60] demonstrates the preoccupation with the geographical and demographic context of outbreaks—where and how a disease spreads appears to matter more than whether it is spreading at all. The 2014 Ebola outbreak is also reflective of such practice wherein a PHEIC was declared only once the virus was seen to pose a threat beyond the three affected countries in West Africa [12,30,61]. Such selective securitisation highlights the complex and often inequitable considerations underlying health emergencies, suggesting that the decision to 'de-securitise' in the case of mpox – to downplay or lift emergency measures- may also reflect a strategic prioritisation of certain populations or regions over others [55].

The mechanism of 'norm-setting' plays an important role in shaping and reinforcing the other four mechanisms as it leads to the establishment of the standards and frameworks that define what is legitimate or acceptable during health crises and beyond – both vertically (the normalisation of practices from the West), and horizontally (across and within governments). Although actors that operate through the mechanism may not overtly acknowledge or may not be consciously aware of the other four mechanisms, the measures they put in place, even when done in compliance with a previously set (health security) agenda or norm, can legitimise health security framing and practices in response to crises. For example, vertically, it can operate through donor-recipient relationships. Most donors (bilateral and/or multilateral), even if tokenistically, make efforts to adhere to principles of sovereignty and country ownership and pledge to align their support with the priorities of countries [62]. However, in practice, partners of certain Global North countries, such as the US, have significant influence in agenda setting and country prioritisation by stipulating which programs or services they will fund and the type of technical support they provide [24,63], which can permeate funding models of other agencies [24,64]. Such practices can, therefore, be observed as part of a bigger security agenda to advance foreign policies and 'soft' power within global health governance [63–65] and greatly influence how specific components of the health system may be (de) prioritised and, by extension, set norms and standards [24,64]. Across and within governments, a national government setting norms at the country level in compliance with international standards can impede the ways in which sub-national governments may respond to their local health needs such that increasing the country's capacity to respond and deal with an infectious disease that is an international and national priority can take precedent over a local health crisis. [66]

The findings of this review reflect the broad literature on health securitisation and address the political roots of its existence and the politicised ways in which it is operationalised. By examining the framing of health issues through a security lens, this literature reveals how the securitisation process may not be neutral or objective but is often shaped by power dynamics, political agendas, and the interests of various actors involved. Acknowledging the power differentials between these actors, the synthesis highlights and reaffirms the notion of how securitisation can disproportionately empower certain stakeholders while disempowering others [18]. By revealing the mechanism through which securitisation operates, this synthesis expands on existing literature that primarily focuses on the international-national interface. The synthesis also surfaces the influence of securitisation dynamics at the national-subnational interface, that is, the centralisation of

power to the 'state' or national government, showing that the mechanisms – the rationale for action – are not limited to only global actors or dynamics globally, but can occur at different levels of governance across and within national governments. It also focuses on the effects on individuals, groups, and nations (in the form of actor 3), which are often at the margins of most analyses.

A potential limitation of this realist synthesis is that the literature we synthesised reflects what appears to be predominantly a body of work that, by addressing the process of securitisation, adopts a critical perspective and stance toward it and may not be fully reflective of the perceptions and applications of actors who use 'health security' in policies and practice in a 'benign' way; that is; detached from its political roots. Two things should be considered in this light: 1, contextual factors always matter, historically and in real-time, such that framing a health issue as a security threat within the securitisation theory is inherently political and does not embody the experiences of the Global South with the same regard [67]; and 2, that even when operationalised on the ground merely as a means to mobilise resources and build health system capacities, there may be other motivations at a higher level of governance, such that both are not mutually exclusive.

Our findings emphasise the need for policies at all levels of governance to attend to the potential consequences and address the complexities of health security framing, including public health professionals who work on the ground, as policymakers and implementers. Our findings offer an opening to exploring the political roots of such framing for public health practitioners who otherwise do not engage in those considerations and highlight the importance of doing so. Recognising how certain individuals, groups and nations can be adversely affected can enable practitioners to reflect on their role as 'norm-setters' even if politics at a higher level is outside their reach. Given the colonial history of global public health and its entanglements with securitisation, there is a responsibility to confront and engage ethically with such complexities.

Realist syntheses go beyond standard systematic reviews and enable the development of C-M-O configurations that uncover causes and link context to outcomes. However, they are not reproducible or standardised and, therefore, potentially subjective based on the judgment and the methodological innovation of the reviewers. A limitation of this review is that primary papers used in the synthesis were not written with underlying mechanisms in mind – and in reporting outcomes, the papers did not explicitly assess prioritisation as a result of health securitisation and only implicitly demonstrated and were therefore abstracted by the reviewers – that is; decisions made, or actions taken by actors described. However, as the realist syntheses are not about "producing definitive facts" [41], the review was able to generate information about how context can impact how health security framing is used through the mechanisms. The C-M-O configurations generated can then be further refined or used as a starting point in empirical investigations of how health security framing manifests on the ground in health systems around the world.

## Conclusions

In summary, we reviewed the literature on health securitisation and identified five mechanisms through which actors securitise responses to emergencies and non-emergencies from 27 high, middle, and low-income countries: 'uncertainty' (reflecting ambiguity of risks), 'self-protection' (reflecting national safeguarding), 'self-reliance' (reflecting the need to be autonomous), 'self-preservation' (reflecting institutional credibility), and 'norm-setting' (reflecting long-term policy shifts). We also identified contextual factors that support or hinder the mechanisms. We confirmed the political nature of the securitisation process. We built upon already existing frameworks by explicitly identifying the three actors involved in the process and showed that it is possible for actors to move from one position to another. Most notably, we surfaced the effect on an impacted group, such as individuals, groups, or nations. As the majority of work in this area is concentrated in the field of international relations, future primary research may consider exploring in greater detail the effects and outcomes of using health security framing in policy and practice. Global public health practitioners may also use the findings of this synthesis to make sense of how the context in which such a framing is applied and the rationale for such application may influence outcomes, especially for marginalised individuals, groups, or nations.

## Supporting information

**S1 Data:** Articles linked to superscripts in the findings section.
(DOCX)

## Author contributions

**Conceptualization:** Delaram Akhavein, Seye Abimbola.

**Data curation:** Delaram Akhavein, Seye Abimbola.

**Formal analysis:** Delaram Akhavein, Seye Abimbola.

**Investigation:** Delaram Akhavein, Seye Abimbola.

**Methodology:** Delaram Akhavein, Seye Abimbola.

**Project administration:** Delaram Akhavein.

**Supervision:** Meru Sheel, Seye Abimbola.

**Validation:** Seye Abimbola.

**Visualization:** Seye Abimbola.

**Writing – original draft:** Delaram Akhavein.

**Writing – review & editing:** Delaram Akhavein, Meru Sheel, Seye Abimbola.

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
