## [Decision Letter · Decision Letter 0]

17 Mar 2025

PGPH-D-24-02972

How health securitisation shapes health system priorities: a realist synthesis

Dear Dr. Akhavein,

Thank you for submitting your manuscript to PLOS Global Public Health. After careful consideration, we feel that it has merit but does not fully meet PLOS Global Public Health’s publication criteria as it currently stands. Therefore, we invite you to submit a revised version of the manuscript that addresses the points raised during the review process.

Two reviewers have provided their comments on your manuscript below. Please review these and provide a point by point response to each comment.

We look forward to receiving your revised manuscript.

Kind regards,

Joanna Tindall, PhD

Staff Editor

Journal Requirements:

1. Please send a completed 'Competing Interests' statement, including any COIs declared by your co-authors. If you have no competing interests to declare, please state "The authors have declared that no competing interests exist". Otherwise please declare all competing interests beginning with the statement "I have read the journal's policy and the authors of this manuscript have the following competing interests:"

2. Please insert an Ethics Statement at the beginning of your Methods section, under a subheading 'Ethics Statement'. It must include:

1) The name(s) of the Institutional Review Board(s) or Ethics Committee(s)

2) The approval number(s), or a statement that approval was granted by the named board(s) 

3) (for human participants/donors) - A statement that formal consent was obtained (must state whether verbal/written) OR the reason consent was not obtained (e.g. anonymity). NOTE: If child participants, the statement must declare that formal consent was obtained from the parent/guardian.

3. Please send separate figure files in .tif or .eps format. Also, remove the figures from your manuscript file but keep the legends.

4. We have noticed that you have uploaded Supporting Information files, but you have not included a list of legends. Please add a full list of legends for your Supporting Information files after the references list.

5. We note that your Data Availability Statement is currently as follows: “The analysis did not use any additional data beyond what is available in the supplementary document attached as a separate file.”

Additional Editor Comments (if provided):

Reviewers' comments:

Reviewer's Responses to Questions

**Comments to the Author**

1. Does this manuscript meet PLOS Global Public Health’s publication criteria ? Is the manuscript technically sound, and do the data support the conclusions? The manuscript must describe methodologically and ethically rigorous research with conclusions that are appropriately drawn based on the data presented.

Reviewer #1: Yes

Reviewer #2: Yes

2. Has the statistical analysis been performed appropriately and rigorously?

Reviewer #1: Yes

Reviewer #2: N/A

3. Have the authors made all data underlying the findings in their manuscript fully available (please refer to the Data Availability Statement at the start of the manuscript PDF file)?

Reviewer #1: Yes

Reviewer #2: Yes

4. Is the manuscript presented in an intelligible fashion and written in standard English?

Reviewer #1: Yes

Reviewer #2: Yes

5. Review Comments to the Author

Reviewer #1: Overall, I think this is a great paper: well written, based on rigorous analysis, on an important topic.

However, I do have a few comments, of which I hope they can be addressed in a review. Before that, some positionality reflections are in order: I am one of those people from the Global North, mentioned in lines 782-792, who use ‘securitisation’ as a means to mobilise international financial transfers for health. It is not an argument I like, but it may be the only argument that works. By the way, I also use it to argue for a more balanced global health governance.

So here are my comments.

1. You use realist evaluation, describing and analysing context-mechanism-outcomes of securitisation. But the focus of the paper is on mechanisms: the contexts are summarized and very little is written about outcomes. Without having read the papers on which the analysis is based, I do not know if the outcomes mentioned in figure 4 are real outcomes or desired outcomes. Would it be possible to clarify that?

2. I am a bit surprised that ‘common interests’ was not identified as a mechanism. Slogans like “No one is safe, until we are all safe” https://www.afro.who.int/sites/default/files/Progress%20report%2021/docs/WHO-AFRO_SPRP_COVID-19_DG-Message.pdf (and papers using that logic) can indeed be considered as a call to self-protection via action abroad, but a more benign interpretation would be that it is a call to behave as ‘citizens of the world’ or something like that. Did that not come out at all or did you choose to put that under ‘self-protection’?

3. With regards to the ‘norm-setting’ mechanism, I think it is important to make a clearer distinction between norm-setting itself and encouraging norm compliance. The International Health Regulations, for example, are international law: the norms are set at the negotiation tables in Geneva. I have not read the papers that are mentioned here (endnotes 45-49) but looking at the titles, they seem more about norm compliance than about norm setting.

Reviewer #2: Summary:

This is a well-written article authored by several global health and health security thought leaders. It explores the mechanisms leading to the securitization of health and health systems (i.e., health security). The authors cogently build upon existing theory—namely the Copenhagen School’s Securitisation Theory—to expand the literature in substantive and useful ways, and identify five mechanisms—uncertainty, self-protection, self-reliance, self-preservation, and norm-setting—by which ‘health security’ framings are adopted. The manuscript represents a work that I anticipate will draw significant interest from a wide audience across several public health subfields. There are, however, several major and minor points that should be addressed before its publication.

Major Issues:

Methods, P4, L134 – The omission of ‘epidemic’ as a search is curious, especially considering that most health security events since 2010 (e.g., Ebola, polio, Zika, mpox) have not been characterized by the WHO as a pandemic, but rather ‘epidemics’ or ‘global epidemics.’ I recognize that this might seem like splitting hairs, but I strongly encourage the authors to consider running another search with ‘epidemic’ to ensure that they did not inadvertently exclude any relevant literature in their review.

Methods, P4, L149 – The authors state that engaging with the concept of ‘securitisation’ was among the inclusion/exclusion criteria but never define how ‘security’ or ‘securitisation’ was conceptualized in their study. This needs to be done to promote transparency—especially considering that, in their own words, realist syntheses ‘are not reproducible or standardized.’ The authors should, at a minimum, strive to leave readers with a better understanding of how they conceptualized security to promote comprehension of what they did and why certain articles were included and/or excluded.

Results/Discussion – The authors note that they consider the international, national, and subnational levels (P3, L103), but there is relatively little discussion of the subnational level. I recommend that they revise the manuscript to include more in-depth discussion of how these mechanisms operate at subnational levels or revise the manuscript to narrow the scope to the international and national levels and keep the bits in the Discussion section (e.g., ‘dynamics at the national-subnational interface’).

Results/Discussion – The authors note that one limitation of their realist synthesis is that it is overly critical of health security framings. Recognizing that the origins of health security are rooted in colonial and military origins and that the past several years have witnessed growing literature and advocacy efforts focused on decolonizing global health, this critical lens is unsurprising. There is value and validity in this literature and these efforts and health security must grapple with its less than honorable origins. The authors do a very good job of highlighting this. Still, while the authors cannot change the articles identified and included in the review, the manuscript is in essence a narrative and currently reads as a very biased one. Consider, for example, the discussion of military force in the Self-Protection section. While there are certainly power dynamics at play between Western and non-Western countries, Western countries are not deploying their militaries to respond to a health emergency in a foreign country without the explicit permission of said country. To do so would be an egregious violation of sovereignty and an act of war. And, while the precise motivations for engaging with public health responses in this way may be obscure, they are surely not entirely rooted in self-preservation. For instance, thousands of military personnel were deployed from Canada, China, France, Germany, the USA, and the UK to support the 2014 Ebola response—a disease that while highly lethal, is not especially contagious. It seems highly unlikely that all of these countries would have been motivated to send their militaries to support an outbreak response for a disease that is not especially contagious (and, therefore, relatively unlikely to pose existential threats to their respective country) and likely less lethal in better resourced settings (and, therefore, not as serious of a threat) in the name of ‘self-preservation.’ Another example comes in the authors’ discussion of a ‘Western nation ‘at war.’ The focus on race is one manifestation, but far from the only, form of this stigmatization. For example, consider how certain sexually transmissible diseases have led to the stigmatization of minority gender groups within Western nations (e.g., HIV, mpox, etc.). Ultimately, I would challenge the authors to provide a much more thoughtful and balanced discussion in the Results and/or Discussion section because, as is, the manuscript currently begs the question of ‘why does the health security framing exist at all?’ And, despite its flaws and unsavory origins, there is, surely, some value in the framing or else it would not be used.

Minor Issues:

Introduction, P3, L104 – The authors provide some context for the ‘realist approach to evidence synthesis’ in the Methods section, but I recommend also adding a brief sentence explaining what the ‘realist approach,’ in the Introduction ideally before the sentence describing what it is well-suited to answer. It’s nice to know that the approach is suited for answering ‘what works, for whom, in what context?’ but if readers are unfamiliar with the method, they are likely to find this statement confusing.

Introduction, P3, L105 – Change ‘question’ to ‘questions’ as there are multiple posed.

Methods, P4, L170 – The authors should provide more detail about the richness and rigour assessments. What did these assessments entail? Were there criteria used to systematically evaluate the richness and rigour of articles?

Methods, P6, Line210 – The Theoretical Framing subsection is very useful for understanding the manuscript and answered many questions I had when reading the article for the first time. I would recommend the authors consider moving the bulk of this, or at least the first 2 paragraphs, to the beginning of the Methods section.

Methods, P7, Line254 – Recommend defining ‘C-M-O’ in the text to facilitate reading.

Results, P8, Line298 – Revise ‘countries’ to ‘geographies’ or remove Sao Paulo from the list (as it is not a country).

Results, P15, Line621 – The JEE is but one tool in a broader IHR M&E framework. The other elements—including the SPAR and NAPHS process—also seem relevant for norm-setting. Recommend that the authors mention these, as well.

Results, P8–16 – Noting the reference style in the manuscript changes.

Results, P8 – The uncertainty mechanism seems particularly relevant for emerging infectious diseases (e.g., COVID-19) or for reemerging diseases in new geographies (e.g., Zika). The authors should highlight this because it contributes, at least in part, to why certain disease outbreaks and public health emergencies are not securitized by national and international actors.

Results – The focus on post-civil war environments is interesting, but I wonder if the authors would consider revising this focus to ‘areas of civil unrest’ or something similar. To me, it seems equally plausible that many of the discussed outcomes could also come before a civil war or in an area experiencing widespread civil strife.

Figure 3 – The authors should consider providing a new title. The current one is not terribly helpful for understanding what is being depicted in the figure.

Figure 4 – This is the article in figure form, and it has immense potential. It is what will be used lecture slides. It is what will be used in policy briefs. And it is, currently, nearly illegible because there is far too much text. The authors should work to reduce how busy this figure is. For example, “Pre-pandemic/emergency environment in which governments are pressured or encouraged to activate a whole-government response to a crisis’ could be shortened to simply read ‘Governments are pressured or encouraged to activate a whole-government response.’ I would encourage the authors to try to shorten text to one line, maximum.

6. PLOS authors have the option to publish the peer review history of their article (what does this mean? ). If published, this will include your full peer review and any attached files.

**Do you want your identity to be public for this peer review?** For information about this choice, including consent withdrawal, please see our Privacy Policy .

Reviewer #1: **Yes: ** Gorik Ooms

Reviewer #2: No

---

## [Decision Letter · Decision Letter 1]

1 May 2025

How health securitisation shapes health system priorities: a realist synthesis

PGPH-D-24-02972R1

Dear Ms Akhavein,

We are pleased to inform you that your manuscript 'How health securitisation shapes health system priorities: a realist synthesis' has been provisionally accepted for publication in PLOS Global Public Health.

Best regards,

Ryan Essex

Academic Editor

Reviewer Comments (if any, and for reference):

Reviewer's Responses to Questions

**Comments to the Author**

1. If the authors have adequately addressed your comments raised in a previous round of review and you feel that this manuscript is now acceptable for publication, you may indicate that here to bypass the “Comments to the Author” section, enter your conflict of interest statement in the “Confidential to Editor” section, and submit your "Accept" recommendation.

Reviewer #1: All comments have been addressed

Reviewer #2: All comments have been addressed

2. Does this manuscript meet PLOS Global Public Health’s publication criteria ? Is the manuscript technically sound, and do the data support the conclusions? The manuscript must describe methodologically and ethically rigorous research with conclusions that are appropriately drawn based on the data presented.

Reviewer #1: Yes

Reviewer #2: Yes

3. Has the statistical analysis been performed appropriately and rigorously?

Reviewer #1: Yes

Reviewer #2: Yes

4. Have the authors made all data underlying the findings in their manuscript fully available (please refer to the Data Availability Statement at the start of the manuscript PDF file)?

Reviewer #1: Yes

Reviewer #2: Yes

5. Is the manuscript presented in an intelligible fashion and written in standard English?

Reviewer #1: Yes

Reviewer #2: Yes

6. Review Comments to the Author

Reviewer #1: no more comments

Reviewer #2: The authors have adequately addressed all of my comments and I look forward to seeing this manuscript published in the literature.

7. PLOS authors have the option to publish the peer review history of their article (what does this mean? ). If published, this will include your full peer review and any attached files.

**Do you want your identity to be public for this peer review?** For information about this choice, including consent withdrawal, please see our Privacy Policy .

Reviewer #1: **Yes: ** Gorik Ooms

Reviewer #2: No
